# Enhancing Factuality in Detailed Image Captioning with LLM-MLLM Collaboration

## Abstract

Multimodal large language models (MLLMs) capable of interpreting images can generate highly detailed and extensive captions, owing to their advanced language modeling capabilities. However, the captions they produce frequently contain hallucinations. Furthermore, our empirical analysis reveals that existing hallucination detection methods are less effective in detailed image captioning tasks. We attribute this to the increasing reliance of MLLMs on their own generated text, rather than the input image, as the sequence length grows. To address this issue, we propose a novel corrector-based method that decomposes a given caption into atomic propositions, evaluates the factuality of each unit, and revises the caption accordingly. Our method is training-free and can be applied in a plug-and-play manner to any captioning model. Additionally, we introduce an evaluation framework and a benchmark dataset to facilitate the systematic analysis of detailed captions. Our experiments demonstrate that existing approaches to improve the factuality of MLLM outputs may fall short in detailed image captioning tasks. In contrast, our proposed method significantly enhances the factual accuracy of captions, even improving those generated by GPT-4V. Finally, we highlight a limitation of VQA-centric benchmarking by demonstrating that an MLLM's performance on VQA benchmarks may not correlate with its ability to generate detailed image captions.

## 1 Introduction

Numerous image captioning methods utilizing deep neural networks (DNNs) have been proposed (Vinyals et al., 2015; Xu et al., 2015). However, they are generally limited to generating short and concise captions, which constrains their broader application in real-world scenarios. For instance, in cases such as assistance for visually impaired individuals, where it is necessary to provide highly detailed descriptions of the scene in front of the user, these methods may not be suitable.

Following the recent success of large language models (LLMs) (Brown et al., 2020), there have been attempts to use not only text but also information from other modalities as input to LLMs. Notably, many studies have explored multimodal large language models (MLLMs) that incorporate visual information (Li et al., 2023a; Dai et al., 2023; Liu et al., 2024b). These models have demonstrated significantly superior performance compared to traditional models in tasks such as visual question answering (VQA) and captioning (Liu et al., 2024a). In particular, MLLMs, leveraging the advanced language capabilities of LLMs, are able to generate much longer and more detailed captions than conventional captioning models. However, these generated captions frequently contain inaccurate information, including descriptions of objects that are not present in the input image (Leng et al., 2024). Such hallucination problems hinder the practical application of MLLMs in real-world settings.

Three major approaches have been recently proposed to improve the factuality of MLLM outputs: (i) Decoding-based methods (Leng et al., 2024) reduce the probabilities of hallucination-related tokens during the model's decoding process without requiring additional training; (ii) Training-based methods (Liu et al., 2023a) further train the models on curated multimodal datasets to ensure they generate only accurate responses; (iii) Corrector-based methods (Zhou et al., 2024) employ a corrector model that detects and either removes or revises hallucinations present in the model's responses.

In this paper, **we propose a novel corrector-based method called Visual Factuality EnhanceR** (V-FactER). Unlike existing approaches that require training a corrector (Lee et al., 2024), V-FactER improves the factuality of detailed image captions by leveraging the collaboration between an LLM

and MLLM, without the need for additional training. Moreover, unlike methods that target specific types of hallucinations (Li et al., 2023b; Zhou et al., 2024), our approach does not pre-define the hallucination types, allowing it to address a broader range of issues. The method proceeds as follows: (i) an LLM decomposes a given detailed caption into atomic propositions; (ii) an MLLM verifies the truthfulness of each atomic proposition based on the corresponding image; and (iii) the LLM revises the caption accordingly. Our design is particularly motivated by the observation that, as the length of a model's response increases, hallucinations generated later in the sequence become more difficult for existing methods (Wang et al., 2023; Zhou et al., 2024) to detect.

Evaluating the factuality of detailed captions is not straightforward. Through experiments, we demonstrate that conventional caption evaluation metrics such as BLEU (Papineni et al., 2002), ROUGE (Lin, 2004), METEOR (Banerjee & Lavie, 2005), and CIDEr (Vedantam et al., 2015), as well as recently proposed methods (Hessel et al., 2021; Petryk et al., 2024), fail to accurately assess the factuality of detailed captions. To address this issue, **we propose a novel GPT-based method for factuality evaluation and validate its effectiveness through experiments**. Even if a caption contains factual information, however, it may still be considered inadequate if it does not sufficiently capture the visual information. To measure the coverage of captions, **we construct a detailed VQA dataset through a collaboration between humans and an AI agent** (Achiam et al., 2023). If a caption fully encapsulates the information of a given image, questions about the image should be answerable accurately using only the caption, without referencing the image itself.

Our experiments surprisingly reveal that methods designed to improve the factuality of MLLMs, which have proven effective in tasks like VQA (Huang et al., 2024), may be ineffective for detailed image captioning tasks that require longer responses. In contrast, V-FactER significantly enhances the factuality of captions and can be applied in a plug-and-play manner to any captioning model. Our experiments further demonstrate that this improvement extends to captions generated by the state-of-the-art closed model, GPT-4V (Achiam et al., 2023). Finally, we highlight an issue with the current VQA-centric benchmarking (Duan et al., 2024) by showing that an MLLM's performance on VQA benchmarks may not correlate with its ability to generate detailed image captions.

In summary, our **contributions** are as follows:

- We demonstrate that existing hallucination detection methods may perform worse as MLLM response length increases, and we propose a method that can circumvent this issue.
- We introduce V-FactER, a method that significantly enhances the factuality of given detailed image captions. V-FactER is a pipeline that leverages a pre-trained LLM and MLLM.
- We propose an evaluation framework and benchmark dataset that overcome the limitations of existing caption evaluation methods and enable the systematic analysis of detailed image captions.
- We show that while existing methods designed to improve the factuality of MLLM responses may be ineffective for detailed image captioning tasks, V-FactER significantly improves their factuality.
- Our experiments demonstrate that current VQA benchmarks fail to reliably capture the potential of MLLMs in real-world applications, such as visual assistants for the visually impaired.

## 2 RELATED WORK

**Multimodal large language models.** LLMs that process inputs from multiple modalities, including text and other types of data, are referred to as multimodal LLMs (Yin et al., 2023a). Among these, *LLMs that handle visual input have been the most actively researched, and the MLLMs discussed in this paper are focused on this category*. Research on these models primarily explores methods for fusing the output of an independent vision encoder into the input of an LLM. The BLIP models (Li et al., 2023a; Dai et al., 2023) align the frozen vision encoder and LLM using a lightweight transformer (Vaswani, 2017) called Q-Former. The trainable input tokens of the Q-Former interact with the output tokens from the vision encoder through cross-attention, transforming them into input tokens for the LLM. The LLaVA models (Liu et al., 2024b;a) use a simple MLP connector to align the vision encoder with the LLM. All output tokens from the vision encoder, passed through the MLP connector, are used as input to the LLM. The vision encoder's parameters remain fixed during the training of the MLP connector and the LLM. Unlike existing MLLMs, the InternVL models (Chen et al., 2024c;b) have demonstrated the effectiveness of increasing the size of both the vision encoder and the vision-language connector. They utilize a 6-billion parameter vision encoder and an 8-billion parameter vision-language connector. The connector is obtained by fine-tuning the

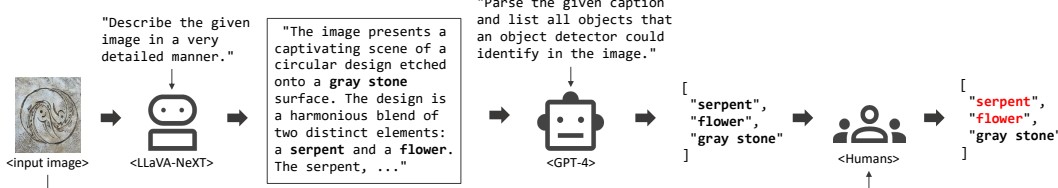

Figure 1: The process of generating a data sample for evaluating the performance of hallucination detection methods in detailed image captioning tasks. Human annotators identify and label object hallucinations within the caption generated by LLaVA-NeXT (Liu et al., 2024a) for an input image.

pre-trained multilingual LLaMA (Cui et al., 2023). Despite the many advancements in open-source MLLMs, closed-source MLLMs such as GPT-4V or GPT-4o[1] still outperform them significantly. As a result, these GPT models represent the upper bound performance in benchmarks and are commonly used to evaluate MLLMs (Petryk et al., 2024). In our work, we demonstrate that captions generated by GPT-4V can be improved using our method, and we use GPT-4o to assess the factuality of captions.

**MLLM hallucinations and mitigation strategies.** MLLMs sometimes generate inaccurate responses. For example, they may incorrectly describe the characteristics of objects in an input image, misrepresent relationships between objects, or even describe objects that do not exist. To mitigate these hallucination problems, decoding-based methods identify factors that induce hallucinations and apply penalties to the probabilities of tokens that are likely to be hallucinations during the decoding process. For instance, VCD (Leng et al., 2024) induces hallucinations using corrupted images, while OPERA (Huang et al., 2024) leverages the correlation between high attention weights assigned to a few summary tokens and hallucinations. Training-based methods focus on exploring training data that can suppress the generation of hallucinations. Liu et al. (2023a) demonstrated that hallucinations can be alleviated by incorporating negative samples—descriptions that explicitly state the absence of certain objects in a given image—into visual instruction tuning datasets. Corrector-based methods (Zhou et al., 2024; Lee et al., 2024) detect, remove, and revise hallucinations present in MLLM responses by using a corrector model. This model is obtained by supervised fine-tuning a pre-trained MLLM. The corrector model then revises the initial response based on the given image.

**Caption evaluation methods.** Since short image captions are relatively easy to obtain reference captions for, we can use matching-based caption evaluation methods (Hossain et al., 2019) to assess them. However, for long and detailed captions generated by MLLMs, the number of reference captions required for such evaluations becomes exceedingly large. Thus, it becomes impractical to evaluate detailed captioning using traditional approaches. Hessel et al. (2021) proposed CLIPScore, a reference-free evaluation method. CLIPScore measures the distance between an image and its caption within the pre-trained joint representation space of CLIP (Radford et al., 2021). Additionally, the authors introduced RefCLIPScore, which uses both the image and reference captions within that same representation space. Chan et al. (2023) addressed the limitations of matching-based methods by utilizing an LLM. The LLM-based metric they proposed, CLAIR, assigns scores to captions based on reference captions using an LLM. Similarly, ALOHa (Petryk et al., 2024) detects hallucinations by comparing a generated caption and its reference information through the use of an LLM.

## 3 METHOD

In this paper, we propose a new corrector-based method. Corrector-based methods typically detect and remove or revise hallucinations within model responses. Unlike existing approaches, which obtain the corrector model through training, our method employs collaboration between a pre-trained MLLM and LLM. Moreover, in contrast to previous methods that are limited to correcting specific types of hallucinations (Zhou et al., 2024), our approach is free from such constraints. We also propose a dataset and framework for evaluating the detailed image captioning capabilities of an MLLM. Unlike

---

[1]https://openai.com/index/hello-gpt-4o/

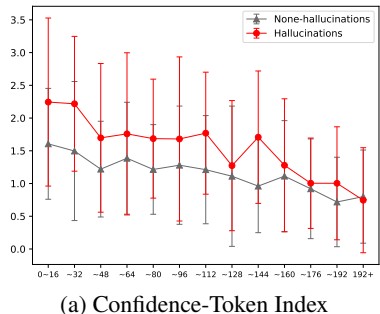 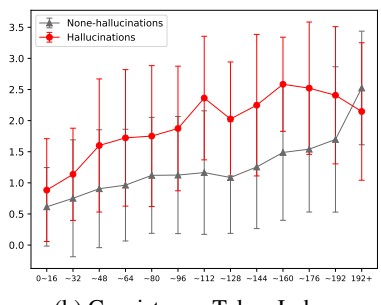

|                          |                          |
| :----------------------: | :----------------------: |
| (a) Confidence-Token Index | (b) Consistency-Token Index |

Figure 2: The hallucination scores of the Confidence and Consistency methods based on object positions within detailed captions. Object hallucinations near the end of the captions (192+) are undetectable by both methods.

existing methods, our proposed evaluation approach allows for the assessment of image captioning models in terms of both factuality and coverage, evaluating each of these aspects separately.

## 3.1 MOTIVATING OBSERVATIONS

Here, we examine the performance of existing hallucination detection methods on tasks that require generating long responses. To facilitate these analyses, we construct a dataset as follows: (i) We prompt an MLLM with "Describe the given image in a very detailed manner." and collect the model's responses for a specified image set; (ii) For the convenience of our analysis, we use an LLM to identify objects that may be hallucinations; (iii) Human annotators then label each parsed object as either a hallucination or not, based on the corresponding image. We use LLaVA-NeXT (Liu et al., 2024a) and GPT-4 as the MLLM and LLM, respectively. Figure 1 illustrates the process of constructing the dataset. To build the dataset, we use a subset of IIW-400 (Garg et al., 2024). We detect object hallucinations using two of the most widely adopted hallucination detection methods:

1. **Confidence** (Zhang et al., 2023; Zhou et al., 2024): This method detects hallucinations using the probability $p_{\mathrm{obj}}$ predicted for the object token when LLaVA-NeXT generates the caption. For multi-token objects, the product of the token probabilities is used. The hallucination score is defined as $H_{\mathrm{obj}} = -\log p_{\mathrm{obj}}$. A higher $H_{\mathrm{obj}}$ indicates a greater likelihood of hallucination.
2. **Consistency** (Wang et al., 2023; Zhao et al., 2024): This method assumes that hallucinations are more influenced by randomness during decoding. Using stochastic decoding, we have LLaVA-NeXT generate 40 detailed captions per image and count the occurrence $t_{\mathrm{obj}}$ of each object in the dataset of Figure 1. The hallucination score is defined as $H_{\mathrm{obj}} = -\log \frac{t_{\mathrm{obj}}}{40}$.

Figure 2 presents the hallucination scores of each method by the position of objects appearing within detailed captions. The horizontal axis of the graphs represents bins of object token indices, with larger token indices indicating positions closer to the end of the caption. The vertical axis represents the mean and standard deviation of the hallucination scores within each bin. Note that Figure 2a reflects the positions and hallucination scores during greedy decoding, while Figure 2b is derived from the average positions and hallucination scores across 40 stochastic decoding iterations. Figure 2 demonstrates that hallucinations generated after the 192nd token are undetectable by the Confidence and Consistency methods. Based on these results, we can infer that existing hallucination detection methods may be ineffective in detecting hallucinations in long detailed captions.

Our hypothesis regarding these results is that *as MLLM outputs become longer, they become more strongly grounded in the text they generate rather than the given image.* In fact, our hypothesis is supported by several recent studies. For example, Liu et al. (2024c) demonstrated that as MLLM responses lengthen, the attention weights assigned to image tokens decrease, and Zhong et al. (2024) showed that MLLM responses are significantly influenced by prior dialogue. Based on this hypothesis, we test a

Table 1: Performance comparison of hallucination detection methods for the dataset of Figure 1.

| Method          | AUROC↑   | FPR95↓   |
| :-------------- | :------: | :------: |
| Confidence      | 57.5     | 95.1     |
| Consistency     | 73.5     | 75.6     |
| Object Detector | 61.5     | 95.7     |
| Isolation       | **81.4** | **71.7** |

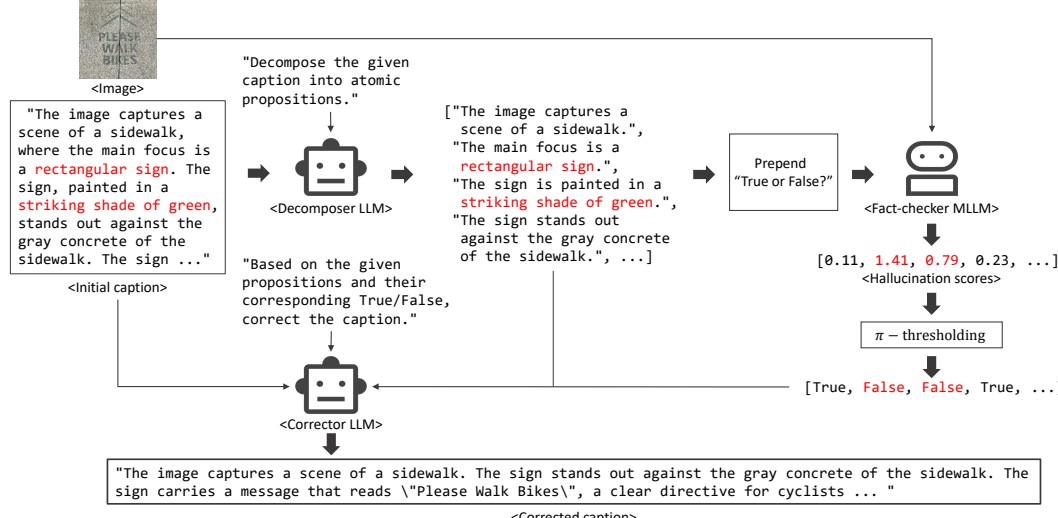

Figure 3: Overview of V-FactER. The decomposer LLM breaks down an initial caption into atomic units. These units are converted into True/False questions and fed into the MLLM along with the image, where each unit is assigned a hallucination score according to Equation (1). Each unit is classified as True or False based on the threshold $\pi$, and the corrector LLM then revises the initial caption based on these results.

method for determining whether each object is a hallucination by disconnecting it from its context (**Isolation**). The Isolation method involves querying the LLaVA-NeXT model with parsed objects using the prompt template, "Is there a {} in the photo?" along with the image. When the probability of the "Yes" token for the object query is $p_{\text{Yes}|\text{obj}}$, the hallucination score is defined as $H_{\text{obj}} = -\log p_{\text{Yes}|\text{obj}}$. We compare the object hallucination detection performance of the Isolation method with that of the Confidence method, the Consistency method, and a method based on an object detector (**Object Detector**) introduced in recent studies (Yin et al., 2023c; Ge et al., 2024). We measure their detection performance on the dataset of Figure 1 using Area Under the Receiver Operating Characteristic (AUROC) and False Positive Rate at 95% true positive rate (FPR95). Table 1 demonstrates that the Isolation method outperforms the others. This suggests that breaking a sentence into smaller units and examining each individually can help detect hallucinations in detailed captions.

## 3.2 VISUAL FACTUALITY ENHANCER

Our motivational observations demonstrate that asking about the presence of objects using a prompt template effectively detects object hallucinations in detailed captions. However, this approach has limitations, as it fails to detect various types of hallucinations. To overcome this limitation, we first decompose each detailed caption into atomic propositions using an LLM. An atomic proposition is a claim or statement that must either be true or false. For example, the caption "A house has a red roof and a chimney" is broken down into "A house has a red roof" and "A house has a chimney." We use an LLM to perform this process, but we allow flexibility in cases where the results do not strictly conform to the definition of an atomic proposition. We then investigate the truth of each decomposed unit using an MLLM. Each unit is converted into a True/False question and independently fed to the MLLM. The hallucination score $H(u)$ for the unit $u$ is defined as follows:

$$H(u) = -\log\left(\min\left(p\left(\text{"True"}|x, Q(u)\right) - p\left(\text{"False"}|x, Q(u)\right), \epsilon\right)\right) \tag{1}$$

$p\left(\text{"True"}\right)$ and $p\left(\text{"False"}\right)$ represent the MLLM's token probabilities for the "True" and "False" tokens, respectively. $x$ and $\epsilon$ denote the input image and a very small constant near zero. $Q(\cdot)$ is a function that converts the input text into a True/False question, which we implement by prepending "True or False?" to the input. Each unit is included in either the True set $\mathcal{T}$ or the False set $\mathcal{F}$, based on its hallucination score. To achieve this, we introduce a hyperparameter $\pi$, such that $\mathcal{T} = \{u|H(u) \leq \pi\}$ and $\mathcal{F} = \{u|H(u) > \pi\}$. Finally, the initial caption, along with the corresponding sets $\mathcal{T}$ and $\mathcal{F}$, is provided to an LLM, which corrects the initial caption to ensure it contains only factual information.

Table 2: Meta-evaluation results across various caption evaluation methods. DOCCI and its synthetic hallucinatory captions are used for the meta-evaluation. The highest-rated caption for each method is highlighted in **bold**.

| Caption | Evaluation Metric | | | | | | | | |
|---|---|---|---|---|---|---|---|---|---|
| | BLEU | ROUGE | METEOR | CIDEr | CLIP-S | RefCLIP-S | CLAIR | ALOHa | Ours |
| Clean | 4.2 | 22.0 | 13.7 | 6.4 | 81.3 | 75.5 | **86.9** | 36.2 | **62.8** |
| Object | **4.9** | **22.3** | **14.5** | 4.8 | 81.0 | 75.3 | 85.2 | 31.5 | 52.3 |
| Attribution | 4.1 | 21.8 | 13.6 | 6.2 | 80.9 | 75.2 | 80.0 | 34.3 | 60.9 |
| Relation | 4.1 | 21.8 | 13.7 | **6.7** | **81.4** | **75.6** | 83.5 | **36.9** | 51.9 |

We name this method, which improves the factuality of detailed image captions through the collaboration of a pre-trained LLM and MLLM, **Visual Factuality EnhanceR (V-FactER)**. V-FactER is training-free and can be applied in a plug-and-play manner to any captioning model. Unlike existing methods that can only address predefined types of hallucinations, V-FactER can detect and correct all hallucinations at the atomic unit level. The pipeline of V-FactER is illustrated in Figure 3.

## 3.3 EVALUATION METHODS

Traditional caption evaluation methods rely on word matching between a predicted caption and its reference captions. This approach works because conventional captioning models generate short captions. However, modern MLLMs produce much longer and more detailed captions, making it impractical to obtain sufficient reference captions for accurate evaluation. Given the enriched content of these image captions, rather than simply evaluating them as good or bad, we aim to assess them systematically by considering two key perspectives:

- **Factuality**: The degree to which the content of the caption is factual and free from hallucinations.
- **Coverage**: The extent to which the caption captures the information contained in the image.

We propose evaluation methods for detailed image captions from these two perspectives.

**Factuality.** If a human were to measure the factuality of a text, it would be natural to decompose the text into units that can be classified as true or false, and then calculate the proportion of true units (Maynez et al., 2020). We adopt this approach to measure the factuality of captions, utilizing the state-of-the-art model GPT-4o. In our framework, GPT-4o decomposes each caption into atomic propositions and determines their truthfulness based on the corresponding image and reference caption. If the number of atomic propositions judged as true and false are $T$ and $F$, respectively, the factuality of the caption is defined as $\frac{T}{T+F}$. This approach enables reliable factuality evaluation using only a single reference caption, unlike conventional methods (Vedantam et al., 2015).

To validate this evaluation method, we use the DOCCI dataset (Onoe et al., 2024), which contains human-annotated detailed image captions. Specifically, for each image in a subset of the dataset, we prepare the following four types of captions (details provided in Appendix B):

1. Clean: The original caption (*e.g.*, *An indoor top-down view captures a white cat with black patches on a wooden floor, attempting to catch a large pale peacock feather flying above it.*).
2. Object: An additional description of an object likely to exist in the image but not actually present is added to the Clean caption (*e.g.*, *An indoor top-down view captures a white cat with black patches on a wooden floor, attempting to catch a large pale peacock feather flying above it. A small red ball is rolling near the cat.*).
3. Attribution: Some object attributions in the Clean caption are modified to be inconsistent with the image (*e.g.*, *An indoor top-down view captures a white cat with black patches on a metal floor, attempting to catch a small dark peacock feather flying above it.*).
4. Relation: Some relationships between objects in the Clean caption are altered to be inconsistent with the image (*e.g.*, *An indoor top-down view captures a white cat with black patches on a wooden floor, attempting to catch a large pale peacock feather flying below it.*).

We evaluate the four types of captions using various image caption evaluation methods (BLEU, ROUGE, METEOR, CIDEr, CLIP-S, RefCLIP-S, CLAIR, and ALOHa), including our own, to determine whether the hallucinations in the three modified types are reflected in the scores. For a fair

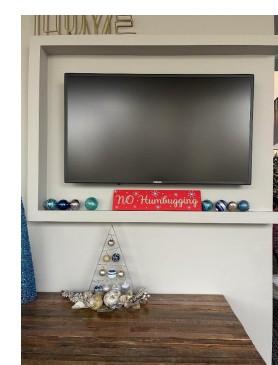

```
1.  What is the main focus of the photo?
    A) A landscape B) A television and decorations C) A group of people D) A building
2.  Where was the photo likely taken?
    A) In a park B) Inside a house C) At a beach D) In a museum
3.  What is the situation depicted in the photo?
    A) A family gathering B) A decorated living space C) A work meeting D) A sporting event
4.  What is in the center of the photo?
    A) A painting B) A television C) A person D) A window

...

47. What is on the left side of the table in the photo?
    A) A lamp B) A blue decorative tree C) A vase D) A stack of books
48. What is the texture of the table in the foreground?
    A) Smooth and shiny B) Rough and rustic C) Soft and cushioned D) Metallic and cold
49. What is in the background of the photo, to the right side?
    A) A kitchen B) A Christmas tree C) A bookshelf D) A window with curtains
50. What type of ornaments are on the triangular decoration on the table?
    A) Animal figurines B) Christmas baubles C) Miniature houses D) Candles
```

Figure 4: An example of our coverage evaluation data sample. The dataset consists of multiple-choice questions with four or fewer options. As demonstrated, the dataset includes questions with varying levels of granularity, ranging from broad to highly detailed. We have an LLM solve these problems using only the provided captions.

comparison, all methods requiring GPT (CLAIR, ALOHa, and ours) use GPT-4o, and all methods requiring reference captions (BLEU, ROUGE, METEOR, CIDEr, CLAIR, ALOHa, and ours) use a separate set (Garg et al., 2024) of human-annotated captions (one reference caption per image).

Table 2 shows that existing metrics are unreliable for evaluating the factuality of detailed image captions. Specifically, BLEU, ROUGE, METEOR, and CIDEr fail to account for hallucinations in the scores and do not assign the highest score to the Clean captions. CLIP can only process up to 77 tokens and operates like a bag-of-words model (Yuksekgonul et al., 2023). This prevents CLIP-based metrics from capturing the full content of detailed image captions, particularly missing Relation hallucinations. ALOHa effectively addresses Object and Attribution hallucinations but fails to capture Relation hallucinations due to its algorithmic limitations. CLAIR detects and reflects all three types of hallucinations in the scores. However, CLAIR does not focus solely on factuality; instead, it allows the GPT model to directly score each caption, applying the evaluation criteria implicitly defined by the GPT model. In contrast, our evaluation method exclusively considers the factuality of the caption. While it does not assign a perfect score to the Clean captions due to GPT-4o's limitations in image understanding, it successfully assigns the highest score to Clean among the four caption sets.

**Coverage.** Even if an image caption contains only factual information, it would not be highly rated if it reflects only trivial aspects of the image. To assess the coverage of captioning models, we propose a QA-based metric and a benchmark dataset. Our coverage evaluation method is based on the assumption that *if an image caption fully captures the information in the image, visual questions about that image should be answerable by referencing the caption alone*.

Our goal is to evaluate detailed image captioning models. Therefore, the visual questions for evaluation must include a variety of detailed and nuanced questions about the images. Given the limitations of existing VQA datasets in this regard (Lu et al., 2022; Yin et al., 2023b; Li et al., 2023b; Yue et al., 2024), we construct a new VQA dataset. However, creating a new VQA dataset that includes a variety of detailed questions requires substantial labor. To reduce the associated costs, we follow the process outlined below to construct our dataset:

1. Generating more than 50 questions per image in the IIW-400 dataset using GPT-4o.
2. Deduplicating the questions for each image using Sentence-BERT (Reimers & Gurevych, 2019).
3. Instructing human labelers to remove or revise questions that can be answered without specific image information, or that are ambiguous or flawed, making them difficult to answer.
4. Annotating the correct answers to the remaining and revised questions by human labelers.

Our coverage evaluation dataset contains a total of 19,899 multiple-choice questions, with each image averaging 49.8 questions. Although we did not explicitly instruct the GPT model to generate detailed questions, it naturally includes them while generating a large number of questions. We present an example of our dataset in Figure 4. While our benchmark dataset can also be used to assess the visual understanding capabilities of MLLMs, we use it to evaluate the coverage of captioning models by having an LLM answer the questions based on the captions generated by those models.

Table 3: Effectiveness of our proposed method across various captioning models. In the V-FactER column, the LLM represents the decomposer and corrector, while the MLLM represents the fact-checker. Avg. denotes the average of the CLAIR, Factuality, and Coverage results.

| Captioner | V-FactER | | Metric | | | |
|---|---|---|---|---|---|---|
| | LLM | MLLM | CLAIR | Factuality | Coverage | Avg. |
| LLaVA-NeXT-7B | - | - | 68.8 | 59.9 | **47.9** | 58.9 |
| | LLaMA-3-8B | LLaVA-NeXT-7B | 74.1 | 72.2 | 46.9 | 64.4 |
| | GPT-4 | LLaVA-NeXT-7B | **74.6** | **73.4** | 46.2 | **64.7** |
| LLaVA-NeXT-13B | - | - | 70.2 | 62.1 | **48.5** | 60.3 |
| | LLaMA-3-8B | LLaVA-NeXT-13B | **75.5** | 77.9 | 45.8 | **66.4** |
| | GPT-4 | LLaVA-NeXT-13B | 73.4 | **79.3** | 45.1 | 65.9 |
| InternVL-Chat-V1.5 | - | - | 74.9 | 65.5 | **48.2** | 62.9 |
| | LLaMA-3-8B | InternVL-Chat-V1.5 | **78.2** | **75.9** | 47.3 | **67.1** |
| | GPT-4 | InternVL-Chat-V1.5 | 77.8 | 75.7 | 47.3 | 66.9 |
| GPT-4V | - | - | 82.4 | 77.1 | **53.5** | 71.0 |
| | LLaMA-3-8B | LLaVA-NeXT-7B | 83.3 | 83.3 | 50.8 | 72.4 |
| | LLaMA-3-8B | LLaVA-NeXT-13B | 81.9 | **85.3** | 48.4 | 71.9 |
| | LLaMA-3-8B | InternVL-Chat-V1.5 | **84.6** | 82.1 | **53.5** | **73.4** |

# 4 EXPERIMENTAL RESULTS AND DISCUSSION

## 4.1 EXPERIMENTAL SETUP

We adopt LLaVA-v1.5-7B, LLaVA-NeXT-7B, LLaVA-NeXT-13B, InternVL-Chat-V1.5, and GPT-4V as the models for both captioning and V-FactER's fact-checking. We use LLaMA-3-8B (AI@Meta, 2024) or GPT-4 as the decomposer and corrector LLMs in V-FactER. Our experiments utilize the IIW-400 dataset, which contains 400 images, each accompanied by a highly detailed, hallucination-free caption. These high-quality reference captions enable precise evaluation of the captioning models.

We employ our proposed factuality and coverage evaluation methods, along with CLAIR, all of which use GPT-4o to evaluate the generated captions. To ensure robust evaluation and assess the recall potential of the captioning methods, we summarize the captions (Ge et al., 2024) generated from five different input prompts using LLaMA-3-8B. The only hyperparameter in V-FactER, $\pi$, is determined using a validation set composed of five images, their QAs, and reference captions. This validation set is constructed by sampling five examples from the DCI dataset (Urbanek et al., 2024). The prompt templates used in our experiments are provided in Appendix B.

## 4.2 IMPROVEMENT IN THE FACTUALITY OF CAPTIONING MODELS

Our proposed V-FactER exhibits a loose factuality-coverage trade-off depending on the hyperparameter $\pi$. Specifically, as $\pi$ decreases, the threshold for determining factual propositions becomes stricter, leading to more propositions being identified for correction. Consequently, factuality increases while coverage decreases (an ablation study on $\pi$ is provided in Appendix A). We first investigate whether V-FactER can enhance the factuality of various MLLMs while minimizing the reduction in coverage.

Table 3 demonstrates that **V-FactER can significantly enhance the factuality of all tested MLLMs while minimizing coverage loss**. The substantial improvement in factuality, compared to the relatively minor coverage loss in the captioning models, is also reflected in the increased CLAIR scores. **Using a more advanced LLM in V-FactER does not necessarily result in greater performance gains**. When applying V-FactER to the LLaVA and InternVL models, there is minimal difference between the results obtained with LLaMA-3-8B and those with GPT-4. This suggests that the LLM's role in V-FactER is relatively straightforward. **V-FactER can improve detailed image captioning even for the state-of-the-art MLLM, GPT-4V**. It can significantly enhance factuality even when used with MLLMs far less capable than GPT-4V. However, in such cases, there is a considerable loss in coverage, as many visual elements recognized by GPT-4V are identified as hallucinations by V-FactER. With InternVL-Chat-V1.5, V-FactER maintains GPT-4V's coverage while improving

Table 4: Performance comparison between our proposed method and other methods regarding detailed image captioning. Base refers to the default image captioning of LLaVA-v1.5-7B without additional techniques.

| Method | CLAIR | Factuality | Coverage | Avg. |
|---|---|---|---|---|
| Base | 62.1 | 52.8 | 34.3 | 49.7 |
| VCD (Leng et al., 2024) | 59.7 | 44.6 | **39.3** | 47.9 |
| OPERA (Huang et al., 2024) | 59.1 | 53.0 | 34.1 | 48.7 |
| Volcano (Lee et al., 2024) | 63.9 | 53.7 | 37.7 | 51.7 |
| LRV-Instruction (Liu et al., 2023a) | 39.7 | 29.1 | 37.8 | 35.5 |
| V-FactER (ours) | **66.3** | **63.4** | 33.1 | **54.3** |

Table 5: Detailed image captioning and VQA performance of various MLLMs. OpenCompass (Duan et al., 2024) includes MMBench v1.1 (Liu et al., 2023b), MMStar (Chen et al., 2024a), MMMU val (Yue et al., 2024), MathVista (Lu et al., 2024), OCRBench (Liu et al., 2024d), AI2D (Kembhavi et al., 2016), HallusionBench (Guan et al., 2024), and MMVet (Yu et al., 2023). For POPE (Li et al., 2023b), we report the average F1 score across the three categories: adversarial, popular, and random. We report the sum of the perception and cognition scores for MME (Yin et al., 2023b). The best results for each metric are shown in **bold**.

| Model | Detailed Image Captioning | | | | Visual Question Answering | | | |
|---|---|---|---|---|---|---|---|---|
| | CLAIR | Factuality | Coverage | Avg. | OpenCompass | MME | POPE | Avg. |
| InstructBLIP-7B | 57.2 | 44.4 | 30.3 | 43.9 | 31.1 | 1391.4 | 86.1 | 38.4 |
| LLaVA-v1.5-7B | 61.1 | 56.3 | 30.5 | 49.3 | 36.9 | 1808.4 | 86.1 | 44.6 |
| LLaVA-NeXT-7B | 63.8 | 58.5 | 42.2 | 54.8 | 44.7 | 1769.1 | 87.5 | 50.8 |
| LLaVA-NeXT-13B | 64.5 | 62.8 | 43.0 | 56.8 | 47.6 | 1745.6 | **87.8** | 53.1 |
| Idefics2-8B | 58.1 | **85.2** | 13.4 | 52.2 | 53.0 | 1847.6 | 86.2 | 57.6 |
| InternVL-Chat-V1.5 | 72.4 | 67.6 | 46.0 | 62.0 | 61.7 | 2189.6 | 87.5 | 65.9 |
| MiniCPM-V-2.6 | 73.1 | 68.9 | 43.6 | 61.9 | **65.2** | **2268.7** | 83.2 | **68.6** |
| GPT-4V | **82.4** | 78.6 | **52.6** | **71.2** | 63.5 | 2070.2 | 81.8 | 66.4 |

factuality. We additionally provide a qualitative comparison in Figure 5 between LLaVA-NeXT-7B with and without the application of V-FactER (referencing the first two rows of Table 3).

## 4.3 COMPARISON WITH OTHER METHODS

Various methods have been proposed to mitigate hallucinations in MLLMs, and they have primarily been validated on VQA and simple captioning benchmarks. We compare V-FactER with two recent decoding-based methods (VCD and OPERA), one corrector-based method (Volcano), and one training-based method (LRV-Instruction) from the perspective of detailed image captioning. All methods, except for LRV-Instruction, use LLaVA-v1.5-7B, while the LRV-Instruction method employs the MiniGPT-4 model (Zhu et al., 2023), as provided by its authors.

Table 4 shows that the VCD, OPERA, and LRV-Instruction methods are ineffective for detailed image captioning. Ironically, applying VCD significantly reduces the factuality of the LLaVA model while increasing coverage. Volcano yields only slight improvements in LLaVA's captions. However, V-FactER substantially enhances the factuality of the captioning model compared to the other methods. *These results suggest that methods proposed to enhance MLLM factuality should be evaluated not only on tasks requiring short responses, such as VQA, but also on detailed image captioning tasks.*

## 4.4 CONSISTENCY BETWEEN MLLM CAPTIONING AND VQA EVALUATION RESULTS

Currently, MLLM evaluations are primarily conducted on tasks that require only short responses, such as VQA tasks (Duan et al., 2024). However, to assess the potential of MLLMs in real-world applications, such as visual assistants, it is essential to evaluate their detailed image captioning abilities. The ranking of models used in our experiments, including LLaVA-v1.5-7B, LLaVA-NeXT-7B, LLaVA-NeXT-13B, InternVL-Chat-V1.5, and GPT-4V, is consistent across both our captioning evaluation results and widely used benchmarks like MMMU (Yue et al., 2024). However, for instance,

Figure 5: An example of a caption generated by V-FactER, with LLaVA-NeXT-7B as both the captioning and fact-checking model and LLaMA-3-8B as both the decomposer and corrector LLM.

some MLLMs may be optimized for VQA tasks that require only short responses, allowing them to rank highly on common VQA benchmarks, yet their limited image captioning abilities could restrict their practical use. To investigate this, we evaluate the detailed image captioning capabilities of various MLLMs and examine whether their rankings are consistent with their rankings on widely used VQA benchmarks. We adopt InstructBLIP-7B (Dai et al., 2023), Idefics2-8B (Laurençon et al., 2024), and MiniCPM-V-2.6 (Yao et al., 2024) as additional MLLMs for the experiment.

Table 5 presents the evaluation results of MLLMs' responses to the prompt "Describe the given image in a very detailed manner" as well as the performance of these models on various VQA tasks. From these results, we observe that the performance of an MLLM on widely used benchmarks does not necessarily reflect its capabilities in detailed image captioning. Specifically, Idefics2-8B ranks mid-tier among the tested models in VQA tasks but falls into the lowest-performing group in terms of detailed image captioning. Its high factuality but low coverage indicates that Idefics2-8B has been trained to provide short and concise answers; this conclusion remains unchanged even when using Idefics2-8B-Chatty (Laurençon et al., 2024). Despite being a relatively small model, MiniCPM-V-2.6 attracted attention by outperforming GPT-4V on benchmarks. However, our results show that the model significantly underperforms GPT-4V in detailed image captioning. Additionally, we find that the factuality of the captions cannot be reliably predicted from the accuracy of MLLMs on POPE (Li et al., 2023b), which was proposed to evaluate object hallucinations.

*Based on these experimental results, we raise concerns about the current MLLM evaluations that are centered around VQA tasks. We encourage the community to also evaluate MLLMs from the perspective of detailed image captioning in order to showcase their full potential.*

## 5 CONCLUSION

Detailed image captioning tasks are closely linked to critical applications, such as visual assistance for the impaired. Our research aims to assess and enhance the potential of MLLMs in these real-world contexts. We propose V-FactER, a method that improves detailed image captions through the collaboration of a pre-trained MLLM and LLM. In addition, we introduce a framework and benchmark dataset for evaluating the factuality and coverage of captioning models. Our experiments validate the proposed evaluation framework and demonstrate that V-FactER significantly improves the factuality of captioning models. We additionally present the following two key observations:

- Methods designed to improve MLLM factuality, which have been validated primarily on VQA or short captioning tasks, may be ineffective for detailed image captioning and can even reduce the factuality of the backbone model's responses.
- High performance on commonly used VQA-centric benchmarks does not necessarily indicate that the model will excel in detailed image captioning.

These observations raise concerns about the current VQA-centric trend in MLLM evaluation. We encourage the community to evaluate MLLMs and related algorithms not only on VQA tasks but also on detailed image captioning tasks to gain a more comprehensive understanding of their potential.

## REPRODUCIBILITY STATEMENT

The prompt templates used in our proposed V-FactER are provided in Appendix B. The factuality and coverage evaluation codes are included in the supplementary material, along with a subset of our proposed benchmark dataset. The full dataset will be made publicly available soon.

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

# A ABLATION STUDY

Table 6: Effectiveness of our proposed method across various captioning models as a function of $\pi$. In the V-FactER column, the LLM represents the decomposer and corrector, while the MLLM represents the fact-checker.

| Captioner | V-FactER | | | Metric | | |
|---|---|---|---|---|---|---|
| | LLM | MLLM | $\pi$ | CLAIR | Factuality | Coverage |
| LLaVA-NeXT-7B | - | - | - | 68.8 | 59.9 | 47.9 |
| | LLaMA-3-8B | LLaVA-NeXT-7B | 1.0 | 74.1 | 72.2 | 46.9 |
| | LLaMA-3-8B | LLaVA-NeXT-7B | 0.5 | 73.6 | 76.9 | 43.7 |
| | LLaMA-3-8B | LLaVA-NeXT-7B | 0.3 | 72.2 | 76.8 | 40.0 |
| LLaVA-NeXT-13B | - | - | - | 70.2 | 62.1 | 48.5 |
| | LLaMA-3-8B | LLaVA-NeXT-13B | 1.0 | 75.5 | 77.9 | 45.8 |
| | LLaMA-3-8B | LLaVA-NeXT-13B | 0.5 | 74.8 | 79.9 | 42.1 |
| | LLaMA-3-8B | LLaVA-NeXT-13B | 0.3 | 72.6 | 80.5 | 39.6 |
| InternVL-Chat-V1.5 | - | - | - | 74.9 | 65.5 | 48.2 |
| | LLaMA-3-8B | InternVL-Chat-V1.5 | 1.0 | 78.2 | 75.9 | 47.3 |
| | LLaMA-3-8B | InternVL-Chat-V1.5 | 0.5 | 79.0 | 78.8 | 46.0 |
| | LLaMA-3-8B | InternVL-Chat-V1.5 | 0.3 | 77.7 | 81.7 | 42.5 |

Our proposed method features a single hyperparameter, $\pi$, which serves as the threshold for classifying atomic propositions as hallucinations or non-hallucinations. Table 6 presents the effects of V-FactER across various models as a function of $\pi$. The results reveal a loose trade-off between factuality and coverage depending on $\pi$. Specifically, in all tested settings, as $\pi$ increases, factuality tends to decrease while coverage increases.

# B PROMPT TEMPLATES

```
prompt_1 = "Describe the given image in a very detailed manner."
prompt_2 = "Provide a detailed description of the specified image."
prompt_3 = "Elaborate on the details of the image provided."
prompt_4 = "Offer an in-depth description of the given image."
prompt_5 = "Thoroughly describe the features of the specified image."
```

Figure 6: The five prompt inputs used to generate captions in our experiments.

```
system:
I want to verify if the given CAPTION is accurate. To assist with this verification, decompose
the given CAPTION into atomic propositions. All parts of the caption must be broken down into
propositions. The outputs should follow the following format:'1. proposition one\n2.
proposition two\n3. proposition three'. For example, break down 'He is tall, thin, and pale'
into '1. He is tall.\n2. He is thin.\n3. He is pale.'
```

```
user:
CAPTION: {caption}
```

Figure 7: The prompt input for LLaMA-3-8B serving as the decomposer.

```
system:
I want to create a caption that includes only facts. Please help me correct the given caption.
The given caption contain things that are not true. Based on the given FACTS and NON-FACTS
remove the non-factual elements from the caption. Place the revised caption between '###'.
```

```
user:
Caption: {caption}\nFACTS:\n{Non-hallucinations among the atomic propositions}\nNON-
FACTS:\n{n{Hallucinations among the atomic propositions}
```

Figure 8: The prompt input for LLaMA-3-8B serving as the corrector.

```
system:
This is a hard problem. Carefully summarize in ONE detailed caption based on the following 5
captions by different people describing the same image. Be sure to describe everything, and
avoid hallucination. Provide the detailed caption in the format '### {Detailed caption} ###'.
```

```
user:
Caption 1: {caption 1st}\n Caption 2: {caption 2nd}\n Caption 3: {caption 3rd}\n Caption 4:
{caption 4th}\n Caption 5: {caption 5th}\n
```

Figure 9: The prompt input for LLaMA-3-8B serving as the summarizer. We use the prompt employed in the work of Ge et al. (2024).

```
if hallucination == "Object"
    prompt_sys = "I want to inject incorrect information into the caption of the given photo.
Your role is to modify about THREE words from the latter part of the given caption that
describe the attributes of the objects so that they do not match the photo."
elif hallucination == "Attribution"
    prompt_sys = "I want to inject incorrect information into the caption of the given photo.
Your role is to imagine an object that isn't actually in the image but could plausibly be
there, and add a very brief part about it to the caption so that they do not match the photo."
elif hallucination == "Relation":
    prompt_sys = "I want to inject incorrect information into the caption of the given photo.
Your role is to change the spatial relationships between the objects so that they do not match
the photo. For example, change 'A person is standing to the right of the car' to 'A person is
standing to the left of the car.' Do not change anything other than the spatial relationships
between the objects."
```

```
system:
{prompt_sys}
```

```
user (presented with the image):
Caption: {caption}
```

Figure 10: The prompt input for GPT-4o used to create the meta-evaluation dataset of Table 2.

```
system:
"I want to use an object detector to check the correctness of an image caption obtained by an
image caption model. Can you help to parse the given CAPTION and list all objects that could
be detected with an object detection model in the image? Please only list the object name and
ignore the description. Please use the name in the CAPTION as it is. Please concatenate them
together with \";\" as separation."
```

```
user:
CAPTION: {caption}
```

Figure 11: The prompt input for GPT-4 used to create the dataset of Figure 1. We use the prompt employed in the work of Ge et al. (2024).

