# OpenReview forum: "Enhancing Factuality in Detailed Image Captioning with LLM-MLLM Collaboration"
_ICLR.cc/2025/Conference — Submitted to ICLR 2025_

### Official Review · Reviewer_J15Y · 2024-10-19

**Soundness:** 2
**Presentation:** 3
**Contribution:** 2
**Rating:** 3
**Confidence:** 4

**Summary:**

1. This work introduces an image caption correction method called **V-FactER** to enhance factuality. The main idea is to break down the detailed caption into atomic propositions with an LLM, verify each with an MLLM, and revise accordingly. The authors find that current hallucination detection methods perform worse as caption length increases, which inspired the proposal of V-FactER.
2. Beyond the method, this work also proposes a new evaluation framework and benchmark dataset to assess caption quality in terms of *factuality* and *coverage*. The authors demonstrate that the new assessment can better reflect various types of hallucinations.
3. This work also suggests that:
    - some current methods aiming to reduce hallucinations in VQA tasks are not effective on captioning tasks;
    - some current hallucination benchmarks focusing on VQA tasks can not capture model performance on captioning tasks.

**Strengths:**

1. **This paper is easy to follow.**
2. **Extensive experiments.** Most of the paper's claims are well supported by experiments. For example, the paper reveals the limitations of the confidence-based and consistency-based hallucination detection methods, which motivate the method; the paper constructs a dataset containing image captions with various hallucinations for comparing different evaluation metrics.
3. **Effective methods.** Experiments demonstrate the effectiveness of the proposed method, which is training-free and works out-of-the-box.

**Weaknesses:**

1. **Moderate Contribution.** The method of decomposing image captions and verifying each component individually is straightforward but lacks novelty. Similar methods have been proposed in prior works, such as FaithScore [1] which utilizes LLMs and MLLMs to extract and verify atomic visual facts, and HalluciDocter [2] which identifies objects, relationships, and attributes in captions and verifies each using specialized models like DINO and BLIP.
2. **Inadequate Baseline Comparison.** It misses out on several methods specifically designed or proven effective for **generation tasks** (rather than VQA tasks) [3], for example, LURE [4], which is also a correction-based method. It's also worth noting that the included baselines like VCD, OPERA, and Volcano do not employ external knowledge, unlike the proposed V-FactER.

[1] FaithScore: Fine-grained Evaluations of Hallucinations in Large Vision-Language Models (arXiv:2311.13614)
[2] HalluciDoctor: Mitigating Hallucinatory Toxicity in Visual Instruction Data (arXiv:2311.13614)
[3] Hallucination of Multimodal Large Language Models: A Survey (arXiv:2404.18930)
[4] Analyzing and Mitigating Object Hallucination in Large Vision-Language Models (ICLR 2024)

**Questions:**

The paper is clear. I have no questions.

---

### Official Review · Reviewer_18Ge · 2024-10-30

**Soundness:** 2
**Presentation:** 3
**Contribution:** 2
**Rating:** 3
**Confidence:** 5

**Summary:**

This paper proposes an image captioning refinement framework called V-FactER, which utilizes a corrector-based approach to decompose and separately evaluate captions. Additionally, the paper introduces a detailed captioning evaluation framework and a benchmark designed to assess the quality of detailed captions with respect to factuality, demonstrating higher validity compared to traditional image captioning metrics such as BLEU, CIDEr, and METEOR. Moreover, the authors present a comprehensive VQA dataset to evaluate the coverage of captions. Qualitative experimental results indicate that V-FactER effectively reduces hallucinations in generated captions.

**Strengths:**

1. This paper addresses the measurement of factuality and coverage in image captions, a topic that has not been extensively explored before.
2. The V-FactER framework appears robust and effective, demonstrating its capability to reduce hallucinations in detailed captions through a method of decomposition and correction of incorrect statements.
3. The authors transform the coverage evaluation of captions into a detailed VQA dataset, allowing for a quantitative assessment of coverage.

**Weaknesses:**

1. The concept of decomposing responses into atomic propositions is not novel. For instance, both FactScore [1], RLAIF-V [2], and FaithScore [3] also utilize the approach of breaking down captions or responses into atomic facts for individual verification. The paper does not mention these papers and discuss the difference.
2. The paper does not provide any information about the correlation between the presented metrics and human judgment, despite human evaluation being crucial when proposing image captioning metrics [4, 5].
3. The motivation for using the IIW-400 dataset is not clear, especially given the existence of numerous other image captioning datasets such as MSCOCO [6] and DOCCI [7]. The authors need to quantitatively explain their reasoning for selecting this particular dataset.

[1] Min, Sewon, et al. "Factscore: Fine-grained atomic evaluation of factual precision in long form text generation." arXiv preprint arXiv:2305.14251 (2023).

[2] Yu, Tianyu, et al. "Rlaif-v: Aligning mllms through open-source ai feedback for super gpt-4v trustworthiness." arXiv preprint arXiv:2405.17220 (2024).

[3] Jing, Liqiang, et al. "Faithscore: Evaluating hallucinations in large vision-language models." arXiv preprint arXiv:2311.01477 (2023).

[4] Chan, David, et al. "CLAIR: Evaluating image captions with large language models." arXiv preprint arXiv:2310.12971 (2023).

[5] Lee, Yebin, Imseong Park, and Myungjoo Kang. "FLEUR: An Explainable Reference-Free Evaluation Metric for Image Captioning Using a Large Multimodal Model." arXiv preprint arXiv:2406.06004 (2024).

[6] Lin, Tsung-Yi, et al. "Microsoft coco: Common objects in context." Computer Vision–ECCV 2014: 13th European Conference, Zurich, Switzerland, September 6-12, 2014, Proceedings, Part V 13. Springer International Publishing, 2014.

[7] Onoe, Yasumasa, et al. "DOCCI: Descriptions of Connected and Contrasting Images." arXiv preprint arXiv:2404.19753 (2024).

**Questions:**

1. How does the V-FactER framework influence the performance on general VQA (Visual Question Answering) datasets? Given that V-FactER aims to refine image captions to be more detailed and free from hallucinations, can you provide insights or evidence on whether this leads to a tangible performance improvement in answering visual questions on standard VQA datasets?


2. Could you provide a broader range of results utilizing various Multimodal Large Language Models (MLLMs) to demonstrate how they perform across the evaluation metrics you proposed, such as CLAIR (Caption Length and Image Relevance), Factuality, and Coverage? This additional data would be immensely valuable in understanding the interplay between a model's inherent capabilities and its performance on these specific metrics.


3. Is there any correlation between the proposed detailed VQA dataset and other general VQA datasets such as VQAv2, GQA, OKVQA, etc.?

---

### Official Review · Reviewer_9DcE · 2024-11-02

**Soundness:** 3
**Presentation:** 3
**Contribution:** 3
**Rating:** 5
**Confidence:** 4

**Summary:**

The authors propose a plug-and-play corrector-based method for multimodal large language models to reduce the hallucinations. The method decomposes a given caption into atomic propositions, evaluates the factuality of each unit, and revises the caption accordingly, by using the large language models. An evaluation framework and a benchmark dataset are also presented to enable the systematic analysis of detailed image captions. Experimental results show the V-FactER can improve the factuality of various MLLMs.

**Strengths:**

1. The proposed method brings improvements for enhancing the factuality and coverage of captioning.
2. The paper is organized well and easy to follow.

**Weaknesses:**

1. The technique innovations of the proposed method are unclear to me. (Zhou et al., 2024) and (Lee et al., 2024) also detect, remove, and revise hallucinations by a corrector model. The proposed method seems to only use an LLM as the corrector.
2. Evaluations with other common metrics such as CHAIR [a] and POPE [b] for hallucinations of captioning are missing. Besides, results on benchmarks about question-answering tasks such as MMBench [c] of the proposed method are also necessary, since the correcting process is also suitable.
3. Qualitative comparisons and analyses about the generated captions are insufficient.

[a] Rohrbach, Anna, et al. "Object hallucination in image captioning." arXiv preprint arXiv:1809.02156 (2018).
[b] Li, Yifan, et al. "Evaluating object hallucination in large vision-language models." arXiv preprint arXiv:2305.10355 (2023).
[c] Liu, Yuan, et al. "Mmbench: Is your multi-modal model an all-around player?." European Conference on Computer Vision. Springer, Cham, 2024.

**Questions:**

See the weaknesses.

---

### Official Review · Reviewer_B57b · 2024-11-04

**Soundness:** 2
**Presentation:** 3
**Contribution:** 1
**Rating:** 3
**Confidence:** 3

**Summary:**

This paper addresses the issue of hallucinations in captions generated by multimodal large language models (MLLMs), which, while capable of producing detailed and extensive image descriptions, often include inaccuracies. The authors identify that current hallucination detection methods perform poorly on detailed image captioning, primarily because MLLMs tend to rely more on their generated text than the input image as the sequence length increases. To tackle this, they propose a training-free, corrector-based method that decomposes captions into smaller propositions, checks their factuality, and revises them as needed.

**Strengths:**

- The paper is well formatted and easy to understand. The proposed solution makes logical sense.

- The proposed method can reduce the hallucination of existing VLM without additional training.

**Weaknesses:**

- The idea of breaking down the output response of an VLM in to series of existence questions (True/False questions) have been explored before in the past (https://arxiv.org/abs/2305.10355). Given this, the proposed method from the authors seem to lack technical novelty for an ICLR paper.

- What is the added computational cost for the proposed method?

**Questions:**

See weaknesses above

---

### Official Review · Reviewer_1WQu · 2024-11-04

**Soundness:** 2
**Presentation:** 3
**Contribution:** 2
**Rating:** 5
**Confidence:** 4

**Summary:**

This paper studies the hallucination problem in detailed image captioning task. Specifically, the authors aim at using corrector based method to find the hallucination of generated caption and modify the hallucination by running a hallucination detection step. The method is training free and can be applied to different MLLMs. Since the method relies on hallucination detection, the authors also notice the limitation of existing methods on long captions and propose a new method that convert hallucination problem into a VQA model and applies an off-the-shelf MLLM. The authors also propose a new evaluation method base don VQA to evaluate detailed image captioning on both factuality and coverage. The authors how their hallucination detection method/evaluation/image captioning method are better than other methods.

**Strengths:**

- The paper is clearly motivated and has a coherent story: convert the detailed image captioning problem into VQA, and apply this concept in both generation and evaluation.
- The authors first examine the existing hallucination detection methods on detailed image captioning and show that using VQA to do hallucination detection is better than other methods. This results justify the reason why we want to use VQA formulation during corrector design as well as in evaluation.
- The proposed method is simple and useful, telling from the experiments.
- The authors also propose an evaluation framework for detailed captioning which is beneficial to the field.

**Weaknesses:**

- Novelty:
    - The vFactER method is very similar to Visual factor checker model.
    - The general concept of problem decomposition and self-criticism is also very common in LLM agent field.
- Different hallucination types:
    - Hallucination detection: when measuring the performance of hallucination detections, the authors only look at object hallucinations.
    - Evaluation: the authors show that the proposed factuality evaluation can rank clean caption over other captions with different hallucinations. However, I think it would be good if the authors can actually add metric decomposition. If it is object hallucination, attribute hallucination or relation hallucination. It would be good to have breakdown for coverage metric as well.
- There is no analysis on the quality of atomic proposition generation.
- The authors didn’t compare their metric to SPICE. SPICE is also kind of coverting captions into some propositions, but in a format of scene graph. SPICE also has the notion and precision and recall which can reflect factuality and coverage. SPICE also have attribute, color etc. breakdowns.
- There are not many V-FactER ablation studies except the choice of difference pi value.
- I think the conclusion from table 5 is not convincing. I feel like detailed image captioning might not be a good benchmark. VQA is a more deterministic task while image captioning might be too ambiguous -- there are many ways to describe an image in a detailed manner. Lower performance on detailed captioning could be just the model are not trained with such caption style and does not mean that the model is not capable. Reporting performance after fine-tuning or few shot in context prompting might be a better indicator.
- The evaluation of the newly proposed factuality metric is using synthetic generated captions. The pattern in the synthesized captions might not match the real hallucination of generated captions in the real world.
- It would be better if there is human evaluation on the generated captions.

**Questions:**

- During factuality computation, the MLLM can get access both image and reference. How does the metric change if only image or only reference caption is provided as input?
- Can the author provide the performance on the coverage VQA dataset without feeding input? Although the authors designed the dataset so that the questions can only be answered with image, it might be still be a nontrivial chance that random guesses are correct based on world knowledge.
- Why the detailed image captioning scores in table 5 do not match the scores in Table 3 (w/ VFactER rows)?
- Will there be loss of information or hallucination when converting caption into atomic propositions and then merge back to a single caption? A encoding + decoding without hallucination check might tell.

---

### Meta-Review · Area_Chair_5QgY · 2024-12-19

**Metareview:**

This paper was reviewed by three experts in the field, and reviewers unanimously agreed to reject the paper. The authors did not provide responses to address the reviewers' concerns.

**Additional Comments On Reviewer Discussion:**

No discussions and changes, as the authors did not provide a rebuttal.

---

### Decision · Program_Chairs · 2025-01-22

Reject